# Karl Barth, Memory, and the Nazi Past: Confronting the Question of German Guilt

## William S. Skiles

Department of General Education, Regent University, Virginia Beach, VA 23464, USA; wskiles@regent.edu

**Abstract:** This article examines Karl Barth's confrontation with the Nazi past in his post-war occasional writings and speeches from 1945 to 1950. My thesis is that as early as January 1945, months before the end of the war in Europe, Barth publicly argued the collective guilt of the German people yet sought not to examine this guilt or demand a "collective punishment"—for the crimes were so great and far-reaching into German society, the responsibility too entangled, that it would be impossible to fully understand or appreciate the crimes committed during this period. Instead, Barth wished simply to acknowledge this guilt, encourage the German people to accept it, and continue with the monumental task of reconstructing Germany. Barth's post-war work proved tremendously influential in challenging the history of the Protestant churches' uncritical stand in obedience to the state.

**Keywords:** Karl Barth; Nazi Germany; coming to terms; post-holocaust theology





The Swiss theologian Karl Barth is best known for his impact on twentieth-century Protestant theology, and particularly his emphasis on the "Christ-event" as God's self-revelation for the reconciliation of the world. But he also worked to put his theology into practice, especially after the destruction of the Second World War. At the war's end, the German people had to make sense of the catastrophe of Hitler's regime and the total conquest and occupation of their nation by the Allied Powers. The German churches began an arduous confrontation with the Nazi past. How did Hitler gain the support of an entire nation, and how did he seduce many German churches to adulterate its gospel message? Any engagement with these questions necessarily raises the issue of German guilt. Even before the war ended, the German churches engaged in a debate over these controversial questions and the struggle for forgiveness and reconciliation. Among the most outspoken and influential churchmen was Karl Barth.[1]

German cities such as Berlin and Dresden lay in total ruins, and the nation's infrastructure was devastated. Estimates are that approximately seven million German soldiers and civilians died among a population of approximately 70 million. Millions of ethnic Germans living in what is today Eastern Europe were compelled to flee and return to Germany. Historians debate the numbers of Germans killed in these expulsions, with numbers ranging from nearly half a million to over two million (Haar 2007, p. 278; Burleigh 2000, p. 799). While the suffering inflicted by Hitler and the Nazis upon the Jews and the peoples of Europe was catastrophic, Germans also suffered greatly as a result of the war.

The emphasis on German suffering was quite common among post-war German leaders, especially church leaders, who argued for the victimization of Germans under the Nazi dictatorship and also the Allied conquest of Germany, thus casting themselves as survivors.[2] Some went so far as to liken the Nazi treatment of the oppositional Confessing Church to the Old Testament's depiction of Israel's suffering under Egyptian and Babylonian persecutions. For example, shortly after the war's end in July 1945, the bishop of Berlin-Brandenburg, Otto Dibelius, delivered a sermon in which he explicitly claimed that "the Lord with his mighty hand has delivered us [the Confessing Church] from the power



of the Devil and led us out of Egypt, out of the house of servitude".[3] As Susannah Heschel has argued, both the oppositional Confessing Church members and the pro-Nazi German Christians appropriated the identity of the Jews in the Hebrew Bible to describe the recent calamity of Germans "who had been liberated from Hitler but conquered by the Allies; having murdered the Jews, the Germans could now take their identity" (Heschel 2008, p. 279). Confessing Church leaders identified themselves with biblical Israel, and in the process exonerated themselves as honorable instruments of God that confronted an oppressive kingdom. Yet, ironically, as the "new Israel" they overwhelmingly failed to speak out for persecuted Jews under the Nazi regime.

When the war ended, Confessing Church pastors contributed to the long process of rebuilding German society. The German churches began to come to terms with the Nazi past immediately after the Second World War, and often Confessing pastors led the conversations (Hockenos 2004, pp. 2, 13–14). As leaders of faith communities they had a unique role in applying faith and scripture to their experiences in Nazi Germany, to the defeat of their nation in war, and to the guilt that would become increasingly apparent as the knowledge of the extent of the Holocaust would become more fully known. Church leaders became important participants in the ongoing dialogue about how to come to terms with the Nazi past and, at the same time, how to shape the historical narrative (Hockenos 2004, p. 3).

As early as 19 October 1945, just months after the end of the war in Europe, the Council of the Evangelical Church in Germany (EKD) issued the Stuttgart Declaration to address the question of German guilt. The council included Confessing Church pastors Hans Asmussen, Hanns Lilje, Otto Dibelius, and Martin Niemöller. Under foreign pressure for the German people to acknowledge their guilt in supporting Hitler and his totalitarian regime, the council affirmed German guilt, and specifically the guilt of the German churches (Hockenos 2004, p. 77). The statement read, in part, "With great anguish we state: Through us has endless suffering been brought upon many peoples and countries . . . We accuse ourselves for not witnessing more courageously, for not praying more faithfully, for not believing more joyously, and for not loving more ardently".[4] The Stuttgart Declaration is evidence that as early as the autumn of 1945, the leadership of the post-war church realized the profound failure of Christian witness, the practice of speaking the truth and power of the gospel to communities of faith. Yet, the statement was only a start in the church's process of coming to terms with the Nazi past. The declaration did not mention the persecution of the Jews or the German churches' specific role in acquiescing or even supporting the Nazi regime (especially many Protestant clergy's support in the early 1930s) (Hockenos 2004, p. 76). The Stuttgart Declaration failed to acknowledge specific sins committed, yet it revealed an acknowledgment that Christians in Germany had morally failed their neighbors and must not only seek forgiveness but change for the better.

While the statement won international approval and proved a positive step forward in reconciliation between Germany and the Allied nations, the German people roundly criticized it, feeling betrayed by its insinuation of collective guilt (Hockenos 2004). What followed was a debate that would last decades about how the church or a people can confess sins, seek repentance, and finally, achieve reconciliation. Coming to terms with the past as a church as an institution became a critical matter of importance for the future of post-war Germany, and clergymen and church leaders took varying positions in their interpretations (Hockenos 2004, p. 74). Conservatives within the EKD created the myth of church resistance and portrayed the Confessing Church as an untainted resistance movement (Hockenos 2004, pp. 47–49). Reformers within the EKD, such as Niemöller, were more critical of the churches' actions in the Nazi past and worked to clarify the nature of the guilt of the church and the pastorate in particular (Hockenos 2004, p. 168).

Even more critical was Karl Barth, the Swiss theologian and one of the foundational leaders of the Confessing Church. As early as a month before the war ended in May 1945, Barth argued that "Germany's most celebrated heroes, institutions, and movements contributed to the rise and positive reception of National Socialism" (Hockenos 2004, p. 57).

Conservatives attacked him for arguing that to rebuild Germany, the Allies had to resoundingly defeat the German nation militarily, impose responsibility for war guilt, and introduce more positive role models that would lead to a healthy society (Hockenos 2004, pp. 59–60). In fact, Barth's theology proved tremendously influential in challenging the history of the Protestant churches' uncritical stand in obedience to the state (Hockenos 2004, p. 123). He even returned to Bonn for a short stay in the summer of 1946 to deliver a series of lectures on dogmatics in the Kurfürsten Schloss.[5] He was first among those forced out of the university under National Socialism to return (Tietz 2021, p. 320). In July 1947, reformers within the EKD, informed and inspired by Barth, gathered in the city of Darmstadt to consider the political failings of Christians in Nazi Germany. Using drafts written by Barth, Hans-Joachim Iwand, and Martin Niemöller, the Council of Brethren of the EKD published the Darmstadt Statement on 8 August 1947 (Tietz 2021, p. 319). This was a statement of guilt for the church's tradition of political conservatism that supported the rise of Nazism. Christians could not give uncritical obedience to the state as if they had no responsibility before God for the actions of the state. This statement did not shift blame onto evil forces, demons, original sin, or God's judgment, as some were wont to do, but rather "assigned culpability to the church's decision to eschew the responsibility that comes with Christian freedom and instead to ally itself with conservative political forces" (Hockenos 2004, p. 128). This statement can be read as an acknowledgment that pastors of the churches, including the Confessing Church, did not sufficiently engage publicly in a critique of the Nazi state.

This article will examine Barth's confrontation with the Nazi past in his post-war occasional writings and speeches from 1945 to 1950 and explore his approach and work toward the rebuilding of Germany after the Second World War. My thesis is that as early as January 1945, months before the end of the war in Europe, Barth publicly argued the collective guilt of the German people yet sought not to examine this guilt or demand a "collective punishment"—for the crimes were so great and far-reaching into German society, the responsibility too entangled, that it would be impossible to fully understand or appreciate the crimes committed during this period. Instead, Barth took a pastoral approach and asked Germans to acknowledge and accept this guilt, to assume civic responsibility and begin the monumental task of reconstructing Germany.

First, I will situate this study in the most relevant contemporary literature on German post-war memory and the debate on the question of guilt (*die Schuldfrage*) for the crimes of the Nazi regime. Next, I will briefly sketch a biography of Karl Barth and his professional and political activities in Germany prior to and during the Nazi regime, as well as his activities in Switzerland after his deportation through to the end of the Second World War. The third section will analyze Barth's evocation of memories of the Nazi past in his post-war occasional writings, that is, his published essays, letters, lectures, and sermons. The fourth and final section will offer conclusions on Barth's post-war confrontation with the Nazi past.

I will examine Barth's post-war occasional writings published or presented between the years 1945 and 1950. His memories of the Nazi past are interspersed throughout his writings and speeches, always with a purpose in service of the Church, the German people, and the task of reconstruction. I will sift out these reminisces and distinguish themes that are important for our study of how Barth came to terms with the Nazi past.

## 1. Historical Context

In the years and decades following the end of the Second World War, the German churches would continue to discuss their complicity in the Nazi past and, through this process, find a path forward to repentance and reconciliation. This is perhaps most evident in the church's debate about antisemitism and its role in the Holocaust. Yet, as Matthew Hockenos argues, only a few church leaders devoted themselves to addressing the church's relation to Jews and Judaism (Hockenos 2004, p. 137). In 1948, the EKD published a document entitled "A Message Concerning the Jewish Question", which rejected antisemitism but retained elements of anti-Judaism. Remarkably, the document further specified the

nature of German (or Christian) guilt. It states, "Christians no longer believed that the promises concerning the Jews still held good; they no longer preached it, nor showed it in their attitude to the Jews. In this way we Christians helped to bring about all the injustices and suffering inflicted upon the Jews in our country".[6] This is an early and explicit acknowledgment that the preaching of the churches failed to portray the Jews as God's covenant people, a failure that contributed to the exclusion, persecution, and extermination of European Jewry. The EKD made reforms, yet ongoing antisemitism in German society and the churches hindered change.

After the war, as Jewish displaced persons and refugees began making their way west into occupied Germany in 1946 and 1947, religious and racial prejudice in the churches were barriers to Christian missionary and aid services (Hockenos 2004, pp. 138–39). The largely unsuccessful post-war efforts of the Jewish missions emphasized converting Jews but still made efforts to engage with Judaism and establish relationships between Jews and Christians (Hockenos 2004, pp. 157–60). Postwar missionary activity in Germany was controversial because of the perception among many Germans that the remaining Jews received preferential treatment. Hockenos even notes that "Parishioners across Germany were so resentful toward remaining Jews . . . that they contested an annual Sunday collection devoted to aid offices for Christians of Jewish descent and Protestant missions that focused on proselytizing among Jews" (Hockenos 2004, pp. 137, 138–52). In this way, the reformers' work in the EKD on coming to terms with the Nazi past and the history of antisemitism in the church was counter-cultural but necessary to reconciliation.

Reformers within the EKD realized the need for further action in repenting for sins against Jews in Nazi Germany. Confessing pastor and theologian Heinrich Vogel of Berlin argued that the Stuttgart Declaration fell far short of offering a clear and explicit statement of repentance (Hockenos 2004, p. 167). Vogel chaired a subcommittee at the Berlin-Weissensee Synod in April 1950 and drafted a document that specifically addressed antisemitism and anti-Judaism in the German churches. The document affirmed the Jewish foundations of Christianity, the fact that the church includes Jewish and Gentile Christians, and that God's covenant with Israel remains unbroken. Moreover, the document stated that the Christians in the German churches have "by omission and silence . . . [become] implicated before the God of mercy in the outrage which has been perpetrated against the Jews by the people of our nation" (Hockenos 2004, p. 199). The statement continues, "We ask all Christians to disassociate themselves from all antisemitism and earnestly to resist it, whenever it stirs again, and to encounter Jews and Jewish Christians in a brotherly spirit".[7] While the statement affirms only the "complacency and not complicity" of the German churches in the Nazi persecution of Jews in Germany, it was the first time a synod of the church repented of sins and repudiated the doctrine of supersession.

## 2. Karl Barth and His Wartime Experience

Before examining Barth's engagement with the legacy of the Nazi period, it is perhaps best to begin by exploring his unique pre-war and wartime experience. Karl Barth was born on 10 May 1886, in Basel, Switzerland, and by the time of his death on 10 December 1968, he was widely acknowledged as the greatest theologian of his time. He is perhaps most well-known for his contribution to Protestant theology, his magnum opus, the *Church Dogmatics*, a thirteen-volume systematic theology published in parts for over thirty-five years.

Barth was raised in a Swiss Reformed household and discovered early in life a passion for theology, which he studied first at the University of Bern, and then in Germany at the Universities of Berlin, Tübingen, and Marburg. After graduation Barth accepted an appointment as the pastor of a parish in the small town of Sifenwil, Switzerland, and though he enjoyed his pastoral work immensely, he decided after ten years' experience to begin his teaching career. He accepted a position at the University of Göttingen and then later at the universities of Münster and Bonn.

As Hitler and the Nazi party gained popularity in the early 1930s, Barth perceived a dangerous movement in German politics and even took a public stand by becoming a member of the Social Democrat Party in May 1931. When Hitler came to power in January 1933, Barth feared for the future of Germany and the German churches. He believed that Hitler presented himself as a false messiah and National Socialism as a false religion to the German people, and he warned his students not to allow Hitler and the Nazi message to encroach on the Christian task of theology and mission (Busch 1975, pp. 223–24).

Barth staunchly opposed Nazism because it introduced confusion and corruption into the church. The Nazi threat to Christianity was no more apparent than in the German Christian movement, a group comprised of mainly Protestant lay people and clergy who, in the words of Doris Bergen, "expected the National Socialist regime to inspire spiritual awakening and bring the church to what they considered its rightful place at the heart of German society and culture" (Bergen 1996, pp. 1–2). In June 1933, Barth publicly criticized the German Christian movement for aligning itself with the Nazi government and adopting a heretical theology that excluded people from the church based on "blood" and "race". The historian John Bowden notes, "What [Barth] was fighting against was essentially the 'German Christian' ideology, which closely identified 'Germanism' with Christianity and saw in the Nazi revolution an act of divine redemption and a source of divine revelation" (Herberg 1960, p. 38). Barth protested that the church must not allow the Nazi government to re-write the message of the gospel and that it must preach Jesus' message "even in the Third Reich, but not under it nor in its spirit" (Busch 1975, p. 226).

By the end of 1933, the German pastor Martin Niemöller instituted the Pastors' Emergency League in opposition to the Nazis and the German Christian movement, which subsequently became the basis for the Confessing Church of Germany. Barth lent his support wholeheartedly to this organization, even drafting a key statement entitled, "Declaration on the Right Understanding of the Reformation Confessions in the German Evangelical Church Today", delivered at the Reformed Synod in Barmen in January 1934. The declaration states: "We reject the false doctrine that the church could and should recognize as a source of its proclamation, beyond and besides this one Word of God, yet other events, powers, historical figures, and truths as God's revelation" (Hockenos 2004, p. 179). Representatives from the three Protestant traditions in Germany (Lutheran, Reformed, and United) accepted this document in protest against the Nazi government and the German Christian movement (Hockenos 2004, p. 23).

In the fall of 1934, Barth's opposition to the Nazi regime intensified. The Nazi government required all civil servants, including university professors, to swear an oath of loyalty to Adolf Hitler. In November 1934, as Professor of Theology at the University of Bonn, Barth refused to swear this oath and was subsequently suspended from teaching (Busch 1975, p. 256). Barth appealed his suspension, yet in the meantime, the Nazi government forbade him even to speak in public. His appeal failed, and by June of 1935, Barth was formally dismissed from the University of Bonn. Immediately upon his dismissal, the University of Basel offered Barth a chair in the theology department, which he accepted gratefully and without delay. By the summer of 1935, Barth returned to his hometown and continued his theological work.

Even in Switzerland Barth opposed the Nazi government in various ways. He chaired the Basel Committee of Swiss Aid commissioned to assist exiled German scholars living in Switzerland (Busch 1975, p. 271). He advocated on behalf of German émigrés for jobs and grants, as well as petitioned church leaders in Europe to assist Jews leaving Germany. Barth continued in his support of the German churches in Nazi Germany by delivering a series of letters to the leadership of the Confessing Church to encourage, advise, and warn in its struggle with the Nazi government (Busch 1975, p. 272). He also raised awareness of the Nazi threat in the Swiss church by publishing articles and presenting lectures that argued that "The trials and the suffering of the church in Germany affect every Swiss who is a conscientious member of his Evangelical Reformed Church just as much as if he were a German citizen" (Busch 1975, p. 275).

When the war broke out on 1 September 1939, Barth took some comfort in knowing that this would spell the end of Hitler and National Socialism. Switzerland declared neutrality only the day before and made clear to the world (and most particularly, Germany) that it would militarily oppose an attack on its territory and call for foreign intervention if necessary (Schwarz 1980, p. 120). But Barth did not believe that Christians should remain neutral in regard to the Nazi threat, and he argued that they should take an active stance against Nazi Germany—if only because of its anti-Christian ideology. Significantly, in April 1940, at the age of 53, Barth enlisted as a soldier in the Swiss armed auxiliary; he refused office duty because he desired to serve his nation as any ordinary Swiss soldier, not as a protected, famous theologian, and he volunteered to stand watch along the Rhine in defense of Switzerland (Busch 1975, p. 305).

It is important to underscore the uniqueness of Barth's perspective. He was the preeminent theologian of his time, a widely known academic and public theologian. He lived in Germany and participated in the social, political and religious issues of the day. He was one of the very few Christian leaders who condemned the Nazi regime from the very beginning, and he was also persecuted by the Nazi regime for his opposition. He was a Swiss theologian who lived in Germany for over twenty years as a student or professor, but importantly, he was a foreigner, an outsider in Germany. He was not German and could not represent Germans in his post-war confrontation with the Nazi past.

Though not a German, he loved Germany and its people, and he identified with them. This is why he had to take a stand against Hitler and his regime. He protested to protect the German churches and the people of Germany; he did this because he loved them. His biographer, Eberhard Busch, reports that on one occasion in 1931, Barth reflected on his feelings about Germany:

> I am well aware of the Swiss element in me, but at the same time want to remain totally and unflinchingly in the center of German theology and the German church . . . And if there is to be talk about my certificate of origin, I cannot think of a better way of showing my love of Germany and my identification with it than by remaining in the heart of Germany, even if I differ from so many Germans by being Swiss. (Busch 1975, p. 217)

Barth was an outsider in Germany, yet in the German churches he found a purpose and a home.

## 3. Confronting the Nazi Past

After the war, Barth had a unique experience of the Nazi period to reflect upon, to learn from, and to share with others. In this section I will examine a few themes in Barth's post-war confrontation with the Nazi period, as evidenced in his occasional writings and speeches from 1945 to 1950; I will explore these memories and analyze his reasons for evoking them. First, it is important to acknowledge that Barth perceived the Nazi past through the prism of the Christian faith, which framed and provided meaning to his memory of the war; he confronted the Nazi past from a decidedly Christian perspective. Second, Barth evoked memories to argue the collective guilt of the German people that they might accept their responsibility and acknowledge their guilt for supporting Hitler and his regime. And third, Barth evoked memories of the Nazi past to encourage the Germans to develop a new and more critical political theology. The ultimate purpose of Barth's use of memory in the post-war period was to help the German people come to terms with the Nazi past, so that he might aid in the reconstruction process.

### 3.1. The Prism of the Christian Faith

As a theologian, it is no surprise that Karl Barth understood the Nazi past in terms of his Christian faith. He understood this past as a profound denial of Christianity, in which Germany embraced a false messiah and a false gospel (Barth 1954, p. 140). In the summer of 1946, Barth delivered a lecture addressed to Christian communities of faith in Düsseldorf, Köln, Bonn, and various other German cities, entitled "The Christian Message in Europe

Today", in which he discussed the diminished glory and dominance of Europe since the end of the Second World War. He spoke as a Swiss man to the German people, but more than this, he spoke as a Christian leader to brothers and sisters in the Christian family, who were all struggling to understand the catastrophe of the Second World War. Here, Barth made an argument based on a distinctly Christian rationale. He wrote,

> What has happened in our day to bring about this great change? It can be explained in a few words: it came about that at the height of European development, here in the heart of Europe, an unparalleled revolutionary movement arose—called the revolution of nihilism; it was however, in reality also the revolution of barbarism, quite simply the revolution of mediocrity. From the Christian point of view it was in its most critical aspect, under the name antisemitism, a revolution against Israel and thereby against the mystery of the incarnation of the Word of God. At all events, it amounted to the taking up of arms, the revolt against everything in Europe that till then had been given the names of justice, order and faith, against everything that had made the European community a great and honored leader in the world. (Barth 1954, p. 168)

Note that Barth understood this revolution as an anti-Christian revolution, not simply non-Christian. He argued that the Nazis waged a "revolution against Israel" and the God who reveals himself. As evidenced in this passage, Barth was determined to travel through Germany and Europe to contend that the cause of European devastation was a turning away from God. Further, he argued that antisemitism was not simply a hatred based on racial or political reasons but a symptom of spiritual degeneration. Fundamentally, Barth understood the Nazi past as a result of a profound, societal-wide spiritual failure.

Barth occasionally evoked memories of the recent Nazi past and compared them with cultural memories of the Judeo-Christian past, thus infusing contemporary events with profound religious meaning. In a lecture entitled "The Germans and Ourselves", presented in Switzerland in January and February 1945, Barth prepared for the imminent end of the Second World War and encouraged his fellow citizens to stand ready to serve a devastated Germany. He gently criticized his own nation's neutrality in the war and referred to the Swiss as spectators of a monumental tragedy—the great fall of Germany. Upon reflection of the rise of the Nazi regime and the depths to which Germany had plunged, he compared his memory of this recent past with the Judeo-Christian cultural memory of Lucifer's fall from the heavens in the Old Testament. Barth wrote, "There is a text in the Old Testament in which may be recognized almost word for word what is now happening and will happen to Germany". He quoted from the prophet Isaiah, chapter 14, verses 12–15:

> How art thou fallen from heaven, O Lucifer, son of the morning!
>
> How art thou cut down to the ground, which didst weaken the nations!
>
> For thou hast said in thine heart, I will ascend into heaven,
>
> I will exalt my throne above the stars of God:
>
> I will sit also upon the mount of the congregation, in the sides of the North:
>
> I will ascend above the heights of the clouds;
>
> I will be like the most High.
>
> Yet thou shalt be brought down to hell, to the sides of the pit. (Barth 1947, p. 87)

As the prophet Isaiah applied this ancient song of derision to the king of Babylon in his own day, so also Barth applied this song to Nazi Germany. Barth lamented the tragedy of Germany's fall. Though the war was not over and the destruction of Germany was not yet known in full, Barth understood that German cities were being reduced to rubble and its people were suffering greatly. He was a spectator awed at the great depth Nazi Germany had fallen, from a nation bent on European domination to one "brought down to hell".

It is a common practice in the Judeo-Christian tradition to understand contemporary events in terms of biblical frameworks. The historian Yosef Yerushalmi, in his book *Zakhor*,

likens this process to "[pouring] new wine into old vessels", in that new memories are understood and given meaning in the context of cultural memory (Yerushalmi 1982, p. 38; see also Assmann 2005, pp. 7–8). This practice enables an individual and, more importantly, the whole community to interpret how God works and reveals himself in the present day. And significantly, evoking the cultural memory of the past encourages participation in the past, that a community might "somehow be existentially drawn" into the past through a religious sense of identification (Yerushalmi 1982, p. 44).

Understood in this way, it is no surprise that Barth engaged in this traditional practice. Hockenos describes the same practice common among leaders in the post-war Protestant Church of Germany. He writes, "The dominant discourse of the church from 1945 to 1950 was borrowed from the Bible; to assuage present suffering, pastors and theologians invoked the traditional Christian concepts such as redemptive suffering, 'God's righteous judgment,' and 'His unfathomable compassion' . . . " (Hockenos 2004, p. 171). Using discourse rife with cultural memory conveys a long-standing tradition of trust and hope in the will of God even in the worst of times—in this way, the church is comforted. And like the German churchmen, Barth wished to evoke memories of the Nazi past and connect them to cultural memory to draw a lesson, to make sense of what had happened in Germany.

Returning once again to our example, Barth contended that Nazi Germany was like the angel Lucifer, who desired glory, power, and dominion, that which belonged to God alone. As Lucifer is responsible and guilty for his sins against God, so also was Germany. Note that according to this cultural memory God judged Lucifer, and it seems clear that Barth understood the same to be true of Nazi Germany, that God would judge Germany. But again, the tone is important to consider. Barth spoke with awe and sympathy at the fall of Germany; his purpose for connecting this cultural memory with the Nazi past was to encourage his fellow Swiss citizens to stand by no longer as spectators but to step up in Christian service, willing to help a nation in great need.

In this same lecture, "The Germans and Ourselves", Barth encouraged the Swiss to offer friendship to a friendless Germany, to reach out as a neighbor to become instruments of grace and mercy. In this context he evoked another cultural memory from the Christian past, this time from the Gospel of Matthew. Jesus calls to all those who are "weary and heavy laden", burdened with shame and guilt, to come and receive God's boundless mercy, grace, and forgiveness (11:28). Barth took Jesus' words as recorded by the early Christian Church and translated them into the present to make a new offer to the German people: Barth wrote in the voice of Jesus,

> 'Come unto me, you unlikable ones, you wicked Hitler boys and girls, you brutal S.S. soldiers, you evil Gestapo police, you sad compromisers and collaborationists, all you men of the herd who have moved so long in patient stupidity behind your so-called leader. Come unto me, you guilty and you accomplices, who now obtain your deserts, as you were bound to do. Come unto me, I know you well, but I do not ask who you are and what you have done, I see only that you have reached the end and must start afresh, for good or ill; I will refresh you, I will start afresh from zero with you . . . ' (Barth 1947, p. 98)

Barth reminded his audience that God grants new beginnings, that God grants mercy and forgiveness. This conception of forgiveness is central to the Christian message. In his letter to the church in Rome, the apostle Paul reflects on the need of all human beings to receive God's offer of forgiveness of sins, for "all have sinned and fall short of the glory of God" (3:23). No one is perfect, and thus, all are in need of forgiveness. It should also be noted that in the Christian tradition there is no limit to forgiveness—no crime too severe, no sin too often committed. This principle is illustrated in Jesus' answer to Peter, who asks how many times one should forgive his neighbor: "Not seven times, but, I tell you, seventy-seven times" (Matthew 18:22). In other words, Jesus counsels Peter to forgive others whenever they sin against him, regardless of their past transgressions.

Barth relied upon this conception of forgiveness to offer hope to those Germans who may be burdened by tremendous guilt for sins committed in the Nazi period. He vividly

transported Jesus from the past into the present to make his offer of forgiveness to the Germans. And significantly, Barth presented a Jesus who is not concerned with the past, and who does not ask probing questions about responsibility and guilt for sins committed. Barth presented a Jesus who wishes only to start over, to begin "afresh from zero", not to unpack the past and lay bare sins. As David Haddorff argues, "After the end of the war, Barth changed his political message from one of resistance to the Nazi regime to one of forgiveness and reconciliation with Germany; thus, political responsibility had shifted from resistance against evil to helping a neighbor in need" (Haddorff 2004, p. 13). The task was to rise up out of the rubble and to begin the work of reconstruction.

The notion of "zero hour" (*Stunde Null*) occasionally recurs throughout Barth's confrontation with the Nazi past. Barth used this phrase to refer to the moment of Nazi Germany's capitulation to the Allied Powers in the Second World War, and it connotes a radical break with the Nazi past. For example, at the conclusion of his lecture, "The Germans and Ourselves", Barth reminded his audience that God "is mighty and victorious at that very zero point and that it is given to [the Germans] and to us [the Swiss] to meet the hopelessness of their situation". He encouraged the Swiss to meet the Germans where they were and to help them in the reconstruction process. He argued,

> What matters is our attitude to those who can have a future only by beginning from zero. And we for our part must likewise begin from zero, that we may be able to stand by them in this situation. If we have to bend low that is no bad beginning, but a good one, perhaps the only possible one, for standing by those who are laid so low. (Barth 1947, p. 122)

Barth's present-day Jesus does not ask what the Germans have done but only acknowledges that the end has come and that they must start afresh, "for good or ill" (Barth 1947, p. 98). Barth's use of the concept "zero hour" must be understood in the context of the Judeo-Christian tradition, that God redeems and gives new life. The old life must be put aside and relegated to the past, so as not to impede the progress of the present.

Despite the goodwill that Barth invariably showed towards the German people, the concept of "zero hour" is problematic because it presumes a complete break with the past, denying all continuities. The sociologist Jeffrey Olick, in his book entitled *In the House of the Hangman*, contends that the concept of "zero hour" is understood by the German people to connote a complete caesura in time in which "all German trajectories", such as intellectual life, culture, politics, economy, etc., came to a definitive end at one particular point and, subsequently, emerged again as if "redefined from outside", meaning the occupation authorities (Olick 2005, p. 7). Olick notes that Germans may refer to May 1945 as "zero hour", in reference to the Nazi unconditional surrender to the Allied Powers, or the term may refer to the year 1949, when West Germany became an independent federal state. In both cases, the "zero hour" marked an unequivocal new beginning for the German people. Olick also points out the political value of the trope in post-war discourse, "[implying] that the slate was wiped clean, that postwar Germany was to be completely distinct from National Socialist Germany", thus distancing the German people—and its leadership—from the actions of Nazi regime (Olick 2005, p. 135).

Barth published the lecture as a brochure in Switzerland, and it caused some controversy because of its tone. The judgment of one Dr. Vital Gawonsky of Bern gives us an indication of the reception of this speech in Swizerland (Barth 2008, pp. 689–90). In a censorship report to the Swiss Press and Radio Communication Department dated 9 February 1945 he wrote that some Swiss considered the speech too hostile toward the German people, calling German youth "a horde of dangerous warriors" and the people as "the shame of National-Socialist garbage" (Barth 2008, p. 690). But Gawonsky contended that the content clearly demonstrates Barth's "extremely serious efforts to understand the German problem and [his] endeavor to help the German people in their catastrophe" (Barth 2008, p. 690). Barth wanted the Swiss and Germans to be friends, but friendship must be forged without "sentimentality" (Barth 2008, p. 690). Barth's point was to help the Germans become a "free and really responsible people" (Barth 2008, p. 690). In March 1945,

Barth sent the published speech to the book trade section of the Swiss Press and Radio Commnication Department for "wide distribution" (Barth 2008, p. 692).

### 3.2. On Collective Guilt

A second theme that emerges in Barth's confrontation with the Nazi past is his concern to establish the collective guilt of the German people. He did not mean to insinuate that all Germans committed crimes in support of the Nazi regime, but rather that all Germans were responsible in some way for supporting their government through sins of commission or omission, for direct or indirect participation in Nazi crimes, or simply for consenting to Nazi rule (Barth 1947, pp. 34–5). Barth argued that all Germans need to acknowledge the ways in which they are responsible for the actions of the Nazi regime.

Olick's analysis of post-war German society is helpful in understanding the accusations of collective guilt in the early post-war period. He argues that the German people perceived the accusation of collective guilt through many forms: the occupied government policy of denazification, which "formally placed all German adults under suspicion until they could be classified"; the results of U.S. and British opinion polls demonstrating strong anti-German sentiment; and the common occupation government's references to "the Germans", insinuating that all Germans share in equal responsibility and guilt for the actions of the Nazi regime (Olick 2005, p. 181). Though it is beyond the scope of this study to document the charges of German collective guilt, it is clear that the German people themselves felt the unbearable burden of this accusation—and thus a need to address it and evaluate its validity.

Barth was among those who accused the German people of collective guilt. In an essay commissioned by the *Manchester Evening News* in April 1945, entitled "How Can the Germans be Cured?", Barth argued that humankind has always been "ill" and that at this point in history, the German people "seem to be the most seriously ill" (Barth 1947, p. 3). This illness can best be described as a political, moral, and spiritual illness that demands a cure if Germany is ever to recover its social and national health. First, he said, Germany must take responsibility for the Nazi regime. Barth contended that Germans have for so long relied upon strong and authoritative leaders to rule them—such as Bismarck, Wilhelm II, and Hitler—that they have failed to take responsibility for the fate of their country. He wrote,

> The Germans are used to being ruled in this or in that way, from a central point within a hierarchy and to obey any word or command coming from no matter how far. This is one of the traits because of which they suffered for centuries and which became deadly 12 years ago—and from which they must now be freed, whatever the price. Each of them must now learn to think for himself, of "community" and "state" in terms of his own political task and duty, instead of waiting for the command of the third person. The fact that individual responsibility for political situations is alien to them explains why it is so difficult to make them understand that they cannot simply be cleared of all charges brought against the Nazi system and all its consequences, but that they must be held responsible for all that has been done to them and to the rest of Europe. (Barth 1947, p. 7)

Barth offered this harsh criticism of German history to make the point that Germans must now accept responsibility for the crimes of the Nazi system. He wrote that the Allies must compel the Germans to participate in the occupation government, to learn to rule themselves, to appreciate a new system of civil administration and new manners of law and order.

Barth evoked the memory of the Nazi past and also the whole of modern German history to argue a negative "special path" (*Sonderweg*) thesis, which, in the words of Hockenos, contends "that some of Germany's most celebrated national heroes, institutions, and intellectual movements contributed to the rise and positive reception of National Socialism" (Hockenos 2004, p. 57). Barth maintained that Germans had for so long held

their political leaders responsible for the state of their nation that they will have trouble in the post-war period accepting responsibility for themselves.

Barth held Germans collectively responsible for all that Adolf Hitler and the Nazi regime did to Germany and the whole of Europe. No one individual was to blame, and no one was free from blame—all are responsible; all share in the guilt. As Christiane Tietz has argued, "Barth didn't think much of making only the political leadership responsible for the years that had just ended. Every single individual had failed to live up to his responsibility" (Tietz 2021, p. 314). In the essay, "How Can the Germans Be Cured?", Barth confronted the German people with their past but, significantly, he refused to delineate degrees of responsibility or guilt, or distinguish who is more or less guilty because, he argued, no such distinctions can be made in such a society-wide catastrophe.

> It is absolutely necessary that [the German people] should realize in the future, and for a long time, consider their own responsibility for their guilt in the past, as well as the task that lies ahead. The German thought has the fatal tendency of pointing to the actions of other people, of emphasizing the guilt of their accomplices, especially the guilt of outsiders, when their most pertinent thoughts should be the ones concerning their own actions. (Barth 1947, p. 10)

Though Barth argued that the Germans "must consider their own responsibility for their guilt in the past", he did not explicate or examine this guilt. He left that for the Germans. He continued,

> This would mean that, for the time being, they are not called upon to analyze and to criticize the past of others, and that they must not be concerned with the future of others; that there is only one matter of importance for them, which is that, considering the heap of cinders at the outcome and end of their contribution to world history, to date, their actions should be concentrated only on the small territory left to them . . . The only thing they are now supposed to do is to turn to the problem of reconstructing German life—unfortunately under conditions created by the German behavior to date—as well as to the best means of paying for the damage, alas, unquestionably caused by them in the world. From this point of view, a German cure would consist in the admission of the fact that in the near future their opinion will not be required in the wider historical framework. (Barth 1947, p. 11)

Barth generalized about the German people, about their thought, behaviors, guilt and "cure". He argued that the Germans must now face the reality that their nation is no longer an influential power in Europe and that their only concern should be reconstruction. Indeed, true renewal and reconstruction could only happen if Germans took responsibility (Tietz 2021, p. 315).

Barth concluded that "all Germans failed to a certain extent—not only some of them, not only this one or that one, because they allowed things to go as far as they have gone" (Barth 1947, p. 14). He did not accept the claim by many Germans that only a few were guilty, that only a few criminals hijacked the German nation and plunged it into a devastating war. And though he did not offer the Germans a guideline on how to examine their guilt, he certainly made it clear that not only should the individual investigate his or her own guilt, but also that German society, as a community, should engage in this exercise together. Barth asked Germans to engage in a sensitive and very complex introspection, yet he did not recommend the manner in which this should be done.

The highly respected and admired German philosopher, Karl Jaspers, entered the debate about the question of German guilt in 1947, the same year that Barth published his controversial essay, "How Can the Germans Be Cured?" In a series of lectures that he later published under the title, *The Question of German Guilt*, Jaspers challenged his fellow German citizens to consider whether and how each German citizen may be guilty of the crimes that had taken place during the Nazi regime. He displayed a remarkable sensitivity toward the German people yet encouraged them to confront their past for the

sake of discovering the truth about their culpability. The result is a nuanced and incisive examination of the nature of German responsibility and guilt during the Nazi period.

Jaspers distinguished four types of guilt and elaborated on the jurisdictions and consequences appropriate to each.[8] The first type is criminal guilt, which results from an individual's violation of a law in his or her society; the court has the sole jurisdiction to deliver a suitable judgment as a consequence. The second is political guilt, which is based on the responsibility of all citizens for the actions of their government; in the case of war, the only jurisdiction belongs to the victor, who may as a consequence exact reparations and a loss of power and rights. The third is moral guilt, which is based on an individual's moral responsibility for his or her own actions; jurisdiction belongs to the conscience alone, and the consequences are penance and moral renewal. The fourth type is metaphysical guilt, which derives from the co-responsibility of each person for the well-being of all other human beings; jurisdiction belongs to God alone, and the consequences for this guilt is a humbling transformation of the conscience before God.

Jaspers concluded that Germans share in collective political guilt, and thus are liable for the actions of the Nazi regime. He ruled out the possibility that all Germans could share in a collective criminal or moral guilt, for these categories are only applicable to individuals. Yet, he asked each German to consider his or her own metaphysical guilt. He wrote, "We [as individuals] should question ourselves, should pitilessly analyze ourselves: where did I feel wrongly, think wrongly, act wrongly—we should, as far as possible, look for guilt within ourselves, not in things, nor in the others; we should not dodge into distress . . . In doing so we face God as individuals, no longer as Germans and not collectively".[9] In the end, only the individual may accuse him- or herself of metaphysical guilt. Jaspers' lectures demonstrate that immediately after the war, Germans were in fact engaging the debate about the question of guilt.[10]

In the essay, "How Can the Germans Be Cured?", Barth does not closely examine German guilt, or distinguish degrees or types of guilt, unlike Jaspers. The result is a rather straight-forward conception of German guilt that lays the responsibility of the Nazi crimes squarely on the German people. Not surprisingly, many Germans were offended, including one man who wrote Barth a letter criticizing his argument as unsophisticated, noting that no person is qualified to pronounce such a judgment on an entire people.[11] This position assumes that all Germans were involved in committing crimes on behalf of the Nazi state. In response to this man's letter, Barth admitted,

> I am not so much concerned with guilt in itself, or collective guilt. I am very much in favor of the Germans, and I mean all the Germans, admitting their responsibility for all which happened since 1933. And by this I do not mean so much, the crimes committed as the road that led and had to led to those crimes. Comparatively few Germans must have taken part in the crimes themselves. But they all took the road leading to these crimes, either in the form of actions or negligence, of direct or indirect participation, of explicit or tacit consent, of unequivocal, active or 'pro-forma' party membership, of political indifference or in the form of all kinds of political errors and miscalculations. How else could the 'small minority of criminals' triumph and National Socialism make world history? (Barth 1947, pp. 34–35)

Barth placed responsibility on the "law-abiding citizens" of Nazi Germany, those who legally and morally supported Hitler's regime. Like Jaspers, he called Germans to accept their political guilt for supporting the Nazi regime and its militarist and racist policies. Yes, there were a few "gangsters" who betrayed the German nations, such as Hitler, Goebbels, Himmler, and other Nazi officials, but this betrayal would never have happened were it not for the support of millions.

Barth lived and worked in Nazi Germany from 1933 to 1935 and witnessed firsthand how his colleagues failed to stand up against Hitler and the Nazis. He singled out for reproach:

> all German professors and other members of the university faculties, mainly
> German nationalists, but also liberals and democrats, whose capitulation and
> conversion [he] had the opportunity to witness with [his] own eyes in 1933. They
> and the German judges, civil servants, ministers, authors, artists, etc., who chose
> to go the way of all flesh ... (Barth 1947, p. 37)

This is as far as Barth ever comes to listing the culpable members of Nazi Germany. Now
that the war was over, Barth called the German people as a whole to account for their
betrayal. He asked for an acknowledgment of guilt and repentance, a turn to rebuilding a
society on a foundation of justice and peace. Yet it is worthy of note that Barth did not ask
each German, unlike Jaspers, to consider their moral and metaphysical guilt, to examine
their consciences and what they owed their neighbors amid the tyranny of the Nazi regime.
Such an examination would take years and decades. But Barth was most concerned in
his speeches and occasional writings in this period with the practical and public reform
needed to rebuild Germany.

Also in response to the essay "How Can the Germans be Cured?", one German man
complained to Barth about Allied acts of cruelty committed in the conduct of the invasion
and occupation of Germany, crimes he compared to Nazis atrocities.[12] Yet, Barth drew a
clear line and refused to morally equate Nazi extermination with the Allied strategy to
win the war in Europe. He argued that because Germans elected Hitler, they must accept
the responsibility for waging and suffering a "total war" (Barth 1947, p. 53). He wrote in
response,

> The fact that this tragedy cost so many German lives is indeed deplorable. I,
> however, ... do not think that this should be called murder, nor that the use of
> block-busters should be at all compared with Oradour and Auschwitz. In spite of
> all the sympathy we have for the German victims, we simply cannot admit that
> the annihilation of the peasants of Oradour and of the Jews in Auschwitz falls
> into the same category with the bombardment of the German industry and the
> communication centers in the interest of winning the war by trying to break the
> impetus of attack and resistance in a nation mobilized for total war.[13]

We can surmise that Barth believed the crimes at Oradour and Auschwitz evince such
unimaginable inhumanity and cruelty that they cannot be compared with the destruction
caused by the strategies and tactics employed in waging modern war. Barth was careful
not to disregard or dishonor the suffering people of Germany, but he felt he must clarify
the ways in which Germans must take responsibility for what had happened in Europe.

Barth provided guidance on how his fellow Swiss citizens should understand and
serve with Germans in the post-war period, as evidenced in the aforementioned lecture
"The Germans and Ourselves", delivered in Switzerland in the winter of 1945. Early in the
essay he reflected on the consequences of Nazi brutality on the future of Swiss and German
relations. Barth argued,

> It is repugnant to me to rehearse, let alone to expatiate on the endless sequence
> of what the National Socialists and thus the Germans have done. We know well
> enough. And it is overwhelmingly what has been done in Germany itself, and
> later wherever the Germans established their authority, which have alienated us
> from them. (Barth 1947, p. 70)

He found the Nazi crimes so atrocious that he did not wish to list them, expose them, or
bring them out into the open. This is not his purpose. But Barth noticed that these acts had
"alienated" the Swiss from the Germans, as if an "iron curtain" has descended between the
two countries (Barth 1947, p. 71). He acknowledged this great obstacle and encouraged his
audience to move forward in partnership with the German people, to overcome the past
and create a future together. Now the Swiss must make a decision; they must step forward
and extend the hand of friendship to the Germans (Barth 1947, p. 70). Only with help can
the German nation rebuild again.

If Barth's postwar approach to engaging the German people was pastoral in nature, then one might well ask why he did not speak more of confession and atonement. The news of Nazi atrocities and the concentration camps that came to light in 1945 and 1946 were overwhelming, and it took time for the German people to come to terms with the knowledge of the Holocaust. In this early stage, one might wonder how one could begin to atone, what penance would suffice, and what meaning confession would have without clearly identifying and understanding the sins committed. Before confession and atonement could begin, acknowledgment and acceptance of guilt were required. But so also, at the same time, Germans had to rebuild their country. Where does one begin in the process of reconstruction, the clearing of rubble and rebuilding or the confession of sins that caused the catastrophe in the first place? Confession and atonement would come slowly in the years and decades to come. Barth called Germans to accept guilt and repent of their sin and then move on to the task of reform and reconstruction. In terms of the anti-Judaic theology that undergirded Nazi racial antisemitism, it would take the Catholic and Protestant churches decades to work through the prejudices embedded in its theology before they could confess specific sins and take corrective steps to rebuild relationships with Jews (Skiles 2021). As Alexander and Margarete Mitscherlich have argued, the German people had to confront the reality of the Nazi past and to undergo the process of working through their feelings of fear, pain, guilt, and shame (Mitscherlich and Mitscherlich 1975). Yet, the pressures of clearing the rubble, reconstructing society, and rebuilding the economy distracted Germans from the challenge of engaging the past.

*3.3. A New Political Theology*

The third theme in Barth's post-war confrontation with the Nazi past is his concern to advance a new political theology in the church. Due to the great failure of the German churches to oppose Hitler and his regime, in the post-war period Barth closely examined the theology that informed the pro-Nazi faction, the so-called German Christians. Of particular concern was the doctrine of natural theology, which argued that humanity could discover the revelation of God through reason, science, and whatever means are available. Yet, for Barth, revelation only comes from God; humanity cannot arrive at the revelation of God on its own. Barth famously addressed this issue during the Nazi period in the Barmen Declaration of 1934, which the Confessing Church proudly accepted as a protest against Nazism and the pro-Nazi German Christian movement, which advocated a crude natural theology. As Hockenos writes, the German Christians "placed the events of 1933, German history, German blood, and even Adolf Hitler alongside the gospel as revelations of God's will" (Hockenos 2004, p. 25). After the conclusion of the Second World War and the ousting of the German Christians from influence in the German churches, Barth continued his criticism of natural theology and argued that the gospel of Jesus Christ is the revelation of God.

In the summer of 1946, Barth presented a lecture entitled "The Christian Community and the Civil Community" in various German cities, including Berlin, Göttingen, Papenburg and Stuttgart. He examined the distinctive roles of these two communities—the Christian and the civil.[14] In this essay Barth argued that Christ as Lord is the center of the Church and the state, and that if the state were to act in ways contrary to the gospel in its work of justice, peace, and equity, then the Church must become politically active (Busch 1975, p. 339). Barth contended that the Church must not remain politically indifferent as it did during the Nazi period. He summed up his argument thus:

> The tasks and problems which the Christian community is called to share, in fulfillment of its political responsibility, are 'natural', secular, profane tasks and problems. But the norm by which it should be guided is anything but natural: it is the only norm which it can believe in and accept as a spiritual norm, and is derived from the clear law of its own faith, not from the obscure workings of a system outside itself; it is from knowledge of this norm that it will make its decisions in the political sphere. (Barth 1954, p. 29)

The law of faith is to be the measure of Christian political action. After the crisis of the churches in the Nazi period, Barth affirmed that natural theology is not a reliable or informative guide to the church in its approach to the state. He advised the Christian community to take a firm critical stance toward their governments and to pose opposition when faith warrants it.

Barth's message certainly resonated in the German churches. It is important to note that toward the end of the war, and certainly afterwards, those who had once supported the Nazi regime began to realize the need to take a more critical stand toward their government. The famous Lutheran theologian Paul Althaus, once enthusiastic about National Socialism, preached a sermon in January 1943 in which he counseled obedience to the governing authorities only if they honored God's commandments: "Therein consists the deepest value of a state, that it holds itself to these commandments. Every authority which despises and neglects these basic commandments, degrades and dishonors its office".[15] Though it is not possible to say how many post-war Protestant German church leaders experienced this shift in perspective immediately after the Second World War, there is evidence that the German churches began to change their unqualified position of support for governing authorities, as demonstrated most notably, in its Darmstadt Statement of August 1947.[16]

In this same essay, "The Christian Community and the Civil Community", Barth challenged the most famous biblical text on civic responsibility found in the apostle Paul's Epistle to the Romans, which instructs Christians to obey all governing authorities as divinely established institutions (13:1–5).[17] Barth responded,

> The last thing this instruction implies is that the Christian community and the Christian should offer the blindest possible obedience to the civil community and its officials. What is meant is that Christians should carry out what is required of them for the establishment, preservation and maintenance of the civil community and for the execution of its task, because, although they are Christians and, as such, have their home elsewhere, they also live in this outer circle. Jesus Christ is still its center: they too are therefore responsible for its stability. (Barth 1954, p. 24)

No doubt remembering the blind obedience of Christians during the Nazi years, Barth is careful to clarify the Christian's responsibility. Each member of the community of faith is to observe the government and to judge its actions. The Christian must learn to take responsibility and to make critical distinctions "between the better and the worse political form and reality; between order and caprice; between government and tyranny; between freedom and anarchy; between community and collectivism; between personal rights and individualism . . . " (Barth 1954, p. 27). Barth clearly did not believe that the Judeo-Christian scriptures endorse the support of all governing authorities; rather, he contended that each individual must judge for themselves if their governing authority fulfills its purpose of establishing a just, free, and ordered society.

Barth's position on developing a new political theology was not without controversy. In the spring and summer of 1948, Barth journeyed to Hungary and presented numerous lectures and sermons to diverse religious and public groups in support of the Hungarian Reformed Church. He commended the Hungarian church leaders for taking an independent position in relation to the new communist regime. In the climate of political change Barth thought it best for the church to remain neutral that it might appeal to those sympathetic to the new order. It was best to wait and observe the political changes before rendering judgment, yet the church must continue to preach the gospel (Tietz 2021, pp. 324–25). In response to these speeches, the well-known Swiss theologian Emil Brunner wrote an open letter entitled "How can one understand this?" to his friend and colleague, mildly criticizing him for not speaking out against the spread of communism in eastern Europe with the same vehemence he employed against the Nazis.[18] Brunner found it "incomprehensible" that Barth did not change his stance after the Soviet Union interfered in Czechoslovakian

politics in 1947–1948, backing a communist coup (Barth and Brunner 2000, p. 347). Brunner, like many others, wanted Barth to explain himself and state his position clearly.

Barth defended himself in a letter dated 6 June 1948, in which he recalled his experience in Nazi Germany and recounted the reasons for which he opposed the regime. His memory of the Nazi past was vivid, filled with strong language and metaphor, and deserves to be quoted at length:

> Whether the essence of National Socialism consisted in its 'totalitarianism' or, according to other views, in its 'nihilism', or again in its barbarism, or antisemitism, or whether it was a final, concluding outburst of the militarism which had taken hold on Germany like a madness since 1870—what made it interesting from the Christian point of view was that it was a spell which notoriously revealed its power to overwhelm our souls, to persuade us to believe in its lies and to join in its evil-doings. It could and would take us captive with 'strong mail of craft and power.' We were hypnotized by it as a rabbit by a giant snake. We were in danger of bringing, first incense, and then the complete sacrifice to it as to a false god. That ought not to have been done. We had to object with all our Protestantism as though against the evil. It was not a matter of declaiming against some mischief, distant and easily seen through. It was a matter of life and death, of resistance against a godlessness which was in fact attacking body and soul, and was therefore effectively masked to many thousands of Christian eyes. For that very reason I spoke then and was not silent. For that very reason I could not forgive the collaborators, least of all those among them who were cultured, decent and well-meaning. In that way I consider that I acted as befits a churchman. (Barth 1954, p. 115)

Again, it is important to note his distinctive Christian perspective; he evaluated the Nazi period "from a Christian point of view". He told Brunner that National Socialism wielded a great, almost mystical power able to "overwhelm our souls". Barth evoked his memories of National Socialism to argue against Brunner's characterization of communism as an equally "evil" political system. From Barth's perspective, the communist regime in Hungary possessed no such power over the people's souls, and there was no battle between life and death as in Nazi Germany.

Barth concluded that he was right not to speak out against the communist regime during his trip in Hungary in the same way that he did against Hitler and his regime simply because the two situations could be equated. Nazism and communism are two different movements and thus demand different approaches (Barth 1954, p. 114). In reference to Barth's view of Soviet communism and its influence in eastern Europe, Haddorff argues, "[His] controversial 'silence' concerning Soviet communism was rooted in a practical (not ideological) politics that was governed by what was most *practically* beneficial to persons within their communities [emphasis in original]" (Haddorff 2004, p. 14). It should be noted that Barth had long been sympathetic to the socialist perspective. He engaged in activities in support of workers' rights as a pastor in Safenwil, and significantly, while in Germany he joined the Social Democrat Party in 1931 (Busch 1975, pp. 71, 217). It is thus no surprise that he did not outright condemn the communist government in Hungary. Though it may be debated whether he was right or wrong in his position, there is no question that he evoked memories of the Nazi past to evaluate the political problems of his own day.

Barth developed this argument in an essay published in a Berlin journal called *Unterwegs* in 1949, entitled "The Church Between East and West". In this insightful essay Barth contended that both the capitalist West and the communist East deserve criticism, but each for different reasons. The Church should not choose sides, to be for one and against the other. The Church must learn to walk between the East and the West so that it might serve all people. He clearly understood the mission of the Church in the context of the post-war period, in a time of great need and suffering. He wrote,

> [The Church's] task must be to call men back to humanity, and that is its contribution to reconstruction. The Church can only be the Church in this particular time if it remains free to fulfill that task. It can only stand for Europe: not for a Europe controlled by the West or the East, but for a free Europe going its own way, a third way. (Barth 1954, p. 145)

Barth reflected back to the Nazi past and its grave inhumanity and brutality, and he argued that the Church must concern itself primarily with reconstruction. He made it clear that this reconstruction was not a physical rebuilding of infrastructures and institutions but a spiritual renewal, an awakening that would prevent another catastrophe such as they had all experienced. The primary task of the Church in the post-war period was "to call men back to humanity". As Haddorff argues, for Barth "[t]he church's primary task is to be a witness to the Word of God, and remind the state of its need for repentance and its purpose of promoting justice and peace. The Christian stance is one of responsible management and *reform* of the state [emphasis in original]" (Haddorff 2004, p. 22). The Church must be unhindered in the pursuit of this task, neither bound by allegiances to the capitalist West or the communist East, but rather guided by the law of faith.

It is significant that Barth considered the task of reconstruction to be a central mission of the post-war Church. The Church was not to investigate war crimes, set up trial courts, or examine German responsibility or guilt, for there were institutions that could do these things. The unique task of the Church was to reach out to a civilization that had lost its way, that had nearly destroyed itself completely. Barth argued that the unique task of the Church was to present the Germans and all of Europe with hope and the offer of redemption and new life through the gospel.

## 4. Conclusions

In the post-war period Karl Barth often confronted the Nazi past, seldom with the aim to explore and investigate what had happened, but rather with the purpose of reconstructing Germany and even Europe itself. First, Barth confronted the Nazi past from a decidedly Christian perspective; he understood the events of the past through the framework of religion. Second, he evoked memories of the Nazi past to argue for the collective guilt of the German people, not with an agenda to explicate crimes or delineate responsibility, but to encourage the Germans themselves to consider how they bore the burden of the Second World War and the Holocaust, that they might rebuild in a spirit of honesty and humility. Indeed, Barth seemed to suggest that one way to acknowledge Germany's collective guilt was for Germans to conscientiously assume civic responsibility in reconstructing and just and peaceful state. And third, Barth evoked memories to inspire a reconsideration of the Protestant church's political theology and, in particular, to encourage the Christian community to take a more critical stance toward the state.

Barth chose not to share in his various post-war writings and speeches memories specific to his own personal experiences, such as his dismissal from the University of Bonn in 1935 or his work assisting German émigrés in Switzerland. Rather, he chose to share predominantly more general memories that had a wider social, political or religious significance, such as his recollection in his essay, "The Christian Community and the Civil Community", that the Nazi past was a period of "blind obedience" on the part of the German people. This is a non-specific memory that many in his audience could relate to and appreciate. In this way he invited men and women to remember for themselves how they experienced the Nazi past.

As a theologian, Barth understood the importance of coming to terms with guilt—sin must be acknowledged and repented before God and humanity. But as a pastor, he also realized that the Germans faced the practical, everyday challenges of the physical reconstruction of their nation—an arduous task that allowed little respite for contemplating the past. He understood that in the immediate aftermath of the war the Germans' priorities would be centered on rebuilding their lives, buildings, and infrastructure. But he still felt compelled as a Christian leader to encourage the German people to consider their

own responsibility and acknowledge their burdensome guilt, so that they might begin the process of healing and reconciliation.

**Funding:** This research received no external funding.

**Institutional Review Board Statement:** Not applicable.

**Informed Consent Statement:** Not applicable.

**Data Availability Statement:** Not applicable.

**Conflicts of Interest:** The author declares no conflict of interest.

## Notes

[1] Important works on this topic include: (Hockenos 2004; Jaspers 1947; Moeller 2001; Spotts 1973; Busch 1975; Tietz 2021).

[2] For example, see the Stuttgart Declaration of Guilt, October 1945. This document drawn up by the Council of the Evangelical Church in Germany refers to the German people as a "Gemeinschaft der Leiden", a "community of suffering". See (Hockenos 2004, pp. 46–47, 187; Barnett 1992, pp. 210–11; Spotts 1973, pp. 62–69; Niven 2006, pp. 1–21; Moeller 2001, pp. 3–4, 44–48; Schroeder 2013, pp. 9, 40–44).

[3] Quoted in (Hockenos 2004, p. 52).

[4] Quoted in (Hockenos 2004, p. 76).

[5] The lectures from this series have been collected and published as (Barth 1959). Barth returned the summer semester of 1947 as well to lecture on the Heidelberg Catechism.

[6] Quoted in (Hockenos 2004, pp. 150–51).

[7] Quoted in (Hockenos 2004, p. 199).

[8] This paragraph is based on Jaspers' analysis on pages 31–36, in (Jaspers 1947).

[9] (Jaspers 1947, p. 114).

[10] Interestingly, neither Barth nor Jaspers refer to each other in their discussions on this topic.

[11] The author is not named, but a copy of the letter is included in (Barth 1947, p. 21).

[12] Again, the author is not named, but the letter is included in (Barth 1947, p. 42).

[13] (Barth 1947, p. 53); on 10 January 1944, German soldiers massacred 642 French villagers in Oradour-sur-Glane in retaliation for French underground resistance to the German Army.

[14] This lecture was later revised and enlarged into an essay published the same year. See (Barth 1954, pp. 13–50).

[15] Quoted in (Ericksen 1985, p. 112).

[16] Hockenos notes that "at Darmstadt the brethren council [of the German Protestant Church] delineated where the church erred politically and resolved that that path would not be followed again", (Hockenos 2004, p. 118); see also page 193 for a full copy of the statement.

[17] The New King James Version reads: "Let every person be subject to the governing authorities; for there is no authority except from God, and those authorities that exist have been instituted by God. Therefore whoever resists his authority resists what God has appointed, and those who resist will incur judgment. For rulers are not a terror to good conduct, but to bad. Do you wish to have no fear of the authority? Then do what is good, and you will receive its approval; for it is God's servant for your good. But if you do what is wrong, you should be afraid, for the authority does not bear the sword in vain! It is the servant of God to execute wrath on the wrongdoer. Therefore one must be subject, not only because of wrath but also because of conscience".

[18] Emil Brunner, "How can one understand this?", undated but sometime before 6 June 1948, in (Barth and Brunner 2000, p. 347).

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
