# Peer review of "Karl Barth, Memory, and the Nazi Past: Confronting the Question of German Guilt"

_religions, doi:10.3390/rel13100974_

Round 1

Reviewer 1 Report

This is a good article. The aim and the results are clearly stated. Some recent publications absolutely need to be mentioned or used: Die Akte Karl Barth, ed. by Eberhard Busch; David Haddorff, Community, State, and Church. Three Essays by Karl Barth with a New Introduction (including Haddorff's introduction); Christiane Tietz's recent biography of K. Barth; maybe also some of Arne Rasmusson's works. Some crude errors betray a lack of intimate knowledge of 20th century Protestant theology (Brunner was not German but Swiss; Barth did become a German citizen). Barth's decision to return teach in Germany right after the end of the war (which is the basis of his book Dogmatik im Grundriss, Engl. trans. Dogmatics in Outline) should be mentioned somewhere. The recent correspondences published in the Karl Barth Gesamtausgabe (e.g. with Visser 't Hooft, Brunner and others) should also be consulted, ideally. But the article can easily be corrected and improved.

Author Response

Thank you for your helpful comments and suggestions. I have revised the article by engaging with more recent publications, such as Die Akte Karl Barth, ed. By Busch; David Haddorff’s introduction in Community, State, and Church; Christiane Tietz’s biography of Barth; and the Barth-Brunner correspondence volume. Let me know if you have any questions. Thank you.

Reviewer 2 Report

The writing before line 128 is awkward.  There ought to be a better way of clarifying your intention than saying "This article" or "in this essay."  An enumeration of your various intentions might be better

Author Response

Thank you for your helpful comments and suggestions. I have revised the awkward working prior to lines 128, and I have revised the “this article” and “in this essay” language. I have also tried to clarify my intentions for the research. Let me know if you have any questions. Thank you.

Reviewer 3 Report

The author provides an extensive and well-researched review of Karl Barth’s post-World War Two call for Germans to acknowledge their “collective guilt” for the crimes of the Third Reich. As the author notes in detail, as an expression of this act of contrition, Barth chose to focus on German’s “responsibility” of establishing a well-ordered and just post-War polity and society. It is only in the concluding paragraph of the article that author offers an analytical perspective to explain why the Protestant theologian restricted the focus of his post-war counsel to the “acknowledgment of collective guilt” and thereby to assume a collective responsibility of re-rebuilding a humane and just Germany. In one short sentence, the author suggests that Barth’s public pronouncements on Germany’s collective guilt were guided by pastoral considerations rather than theological principle.  Drawing upon the philosopher Karl Jasper’s delineation of four types of guilt – which the author brings with scant comment – one might say that Barth limited his acknowledgement of Germany’s war crimes to “political guilt.” The author argues compellingly that quickened by his sense of pastoral calling Barth deemed it necessary to guide German Protestant laity to adjust to the post-war challenge of reconstructing Germany. Accordingly, one may say, the Germany’s collective guilt was to be acknowledge by assuming civic responsibility. Barth the theologian did not speak, as one would have expected, of confession, atonement, penance, repentance.

But one might ask whether Barth’s theological reticence was solely due to an overriding sense of pastoral duty. Perusal of the post-war entries in his diaries, one notes an abiding anti-Judaism (which, to be sure, is to be distinguished from anti-Semitism). One example should suffice: With utter candor, he expressed his reluctance to read Jewish theological affirmations of faith, presumably penned by the likes of Buber and Rosenzweig.

In brief, I believe that the essay under consideration contribution to assessing Barth’s Post-War teachings would gain great depth by a critical analysis not only of Barth’s theological reticence but also his virtual silence about the National Socialism’s persecution of the Jews, culminating in the Final Solution.  One passing bibliographical remark. The author might wish to consider Alexander Mitscherlich, Die Unflätigkeit zu trauen (1967), which address from a psychoanalytical perspective why Germans by and large resisted “working through” their guilt.

Author Response

Thank you for your helpful comments and suggestions. I have revised the article in response to your comments about elaborating on Barth’s reticence to go beyond calling the German people to acknowledge collective guilt, to discuss specifics of confession and atonement for sins committed. I have also engaged with various authors to add more critical analysis to my argument (I have revised the article by engaging with more recent publications, such as Die Akte Karl Barth, ed. By Busch; David Haddorff’s introduction in Community, State, and Church; Christiane Tietz’s biography of Barth; and the Barth-Brunner correspondence volume.)

Let me know if you have any questions. Thank you.